# Fatty Liver, and Not Visceral Fat, Is More Associated with Liver Fibrosis and Diabetes in Non-Obese Japanese Individuals: A Cross-Sectional Study

**DOI:** 10.3390/life10090175

**Published:** 2020-09-04

**Authors:** Noriyo Urata, Miwa Kawanaka, Ken Nishino, Katsunori Ishii, Tomohiro Tanikawa, Mitsuhiko Suehiro, Takako Sasai, Ken Haruma, Hirofumi Kawamoto, Jun Nakamura, Noriaki Manabe, Tomoari Kamada

**Affiliations:** 1Department of General Internal Medicine 2, Kawasaki Medical School, 2-6-1 Nakasange, Kitaku, Okayama City, Okayama 700-8505, Japan; noriyo-urata@med.kawasaki-m.ac.jp (N.U.); k-nishino@med.kawasaki-m.ac.jp (K.N.); katsunori.ishii@med.kawasaki-m.ac.jp (K.I.); t-tanikawa@med.kawasaki-m.ac.jp (T.T.); m.suehiro@med.kawasaki-m.ac.jp (M.S.); tksasai@med.kawasaki-m.ac.jp (T.S.); kharuma@med.kawasaki-m.ac.jp (K.H.); h.kawamoto@med.kawasaki-m.ac.jp (H.K.); 2Department of Clinical Pathology and Laboratory, Kawasaki Medical School, 2-6-1 Nakasange, Kitaku, Okayama City, Okayama 700-8505, Japan; j.nakamura@med.kawasaki-m.ac.jp (J.N.); n_manabe@odn.ne.jp (N.M.); 3Department of Health Care Medicine, Kawasaki Medical School, 2-6-1 Nakasange, Kitaku, Okayama City, Okayama 700-8505, Japan; tkamada@med.kawasaki-m.ac.jp

**Keywords:** nonalcoholic fatty liver disease, lean nonalcoholic fatty liver disease, visceral fat, non-obese, fatty liver, insulin resistance

## Abstract

Asians are known to be more likely than Westerners to develop fatty liver and lifestyle-related diseases in spite of their weight. However, the relationship between fat accumulation and lifestyle-related diseases in non-obese Asians is unknown. Therefore, this study aimed to analyze visceral fat and hepatic fat in participants with a normal body mass index (BMI) and examine their characteristics during a medical checkup. This cross-sectional study was conducted on 663 of 1142 patients who underwent abdominal ultrasonography and who had an alcohol intake (converted to ethanol) of <30 g/day for males and <20 g/day for females and a BMI of <25 kg/m^2^ during a health checkup. Participants were classified into four groups: group A, visceral fat accumulation (VFA) (−) and fatty liver (FL) (−) (*n* = 549); group B, VFA (+) and FL (−) (*n* = 32); group C, VFA (−) and FL (+) (*n* = 58); and group D, VFA (+) and FL (+) (*n* = 24). The frequencies of lifestyle-related disease complications, liver function tests, and liver fibrosis were evaluated among the four groups. Compared with group A (control), groups B, C, and D had a higher number of males, BMI, abdominal circumference, ALT, AST, γ-GTP, triglyceride, uric acid, fasting blood sugar levels, and incidence of hyperlipidemia. Groups C and D had higher ALT, HbA1c, cholinesterase, and triglyceride levels, FIB4 index, and the number of patients with diabetes mellitus (DM) than groups A and B; however, there was no difference between groups A and B. FL is a risk factor of DM and liver fibrosis in non-obese Japanese individuals; however, VFA only is not a risk factor of DM and liver fibrosis.

## 1. Introduction

Nonalcoholic fatty liver disease (NAFLD) is a common chronic liver disease worldwide [1]. The development of fatty liver disease has been known to be associated with weight gain, hypertension, hypertriglyceridemia, and obesity [2,3,4]. NAFLD was diagnosed by excluding other chronic liver diseases, including “excessive” alcohol consumption. Recently, a positive criterion has been proposed for metabolic dysfunction-associated fatty liver disease (MAFLD). The criteria are based on evidence of hepatic steatosis in addition to one of these criteria: overweight/obesity, type 2 diabetes mellitus (DM), or evidence of metabolic dysregulation [5].

In NAFLD, the body mass index (BMI; kg/m^2^) of Asians is not very high as compared with that of Westerners with NAFLD. The so-called lean NAFLD (BMI, Caucasians ≤ 25 kg/m^2^; Asian ≤ 23 kg/m^2^) has been diagnosed in 10–20% of non-obese Asians with NAFLD [6,7] Asians are known to be more susceptible to BMI-related metabolic disorders compared with Westerners. In non-obese Asians, as in other ethnic groups, visceral fat accumulation (VFA) and fatty liver (FL) are reported to be associated with insulin resistance and metabolic abnormalities [8,9,10,11,12,13]. Metabolically obese, normal-weight Asians have 20–30% lesser insulin sensitivity and 30–40% higher fasting and postprandial insulin secretion than metabolically healthy, lean controls [14] Among Japanese men with BMI of 23–25 kg/m^2^, those with hypertension, hyperglycemia, or dyslipidemia have insulin resistance in the liver but less insulin resistance in the muscle. On the other hand, overweight men with metabolic syndrome have impaired insulin resistance in both the muscle and liver [4]. However, the mechanisms of insulin resistance in non-obese Asians have not yet been fully elucidated. A recent report compared the degree of insulin sensitivity among four groups defined by the presence or absence of VFA and FL in non-obese diabetic middle-aged Japanese men. FL was associated with adipose tissue and muscle insulin resistance; however, VFA alone were not associated with insulin resistance. Subjects with both VFA and FL were as resistant to insulin as those with FL alone [14].

Based on these studies [14], we divided our study patients into four groups based on the accumulation of FL and visceral fat and with normal BMI (<25 kg/m^2^) at the time of a medical checkup. We investigated clinically whether there was a difference in lifestyle-related diseases, blood test values, and liver fibrosis progression depending on the FL and VFA sites. This examined the importance of the fat accumulation site.

## 2. Materials and Methods 

### 2.1. Patients

This cross-sectional study was conducted from January to December 2018 at Kawasaki Medical School General Medical Center. In total, 1142 patients underwent abdominal ultrasonography at an annual general medical checkup at Kawasaki Medical Center, Kawasaki Medical School. Annual general medical checkups and abdominal ultrasonography are requested by individuals who desire comprehensive examinations that permit early diagnosis of illness. Of these, 663 patients had a BMI of <25 kg/m^2^ and alcohol intake (converted to ethanol) of <30 g/day for males and <20 g/day for females. Alcohol consumption was determined based on the NASH/NAFLD guidelines of the Japanese Society of Hepatology and those of MAFLD [5,15]. In this study, individuals with a BMI of <25 kg/m^2^ were considered as non-obese, and NAFLD was defined as an alcohol intake (converted to ethanol) of <30 g/day for males and <20 g/day for females. Patients with hepatitis B virus (HBV), hepatitis C virus (HCV), and hepatocellular carcinoma (HCC) were excluded from the study (Table 1). Patients were classified into four groups: group A (control group), VFA (−) and FL (−); group B, VFA (+) and FL (−); group C, VFA (−) and FL (+); and group D, VFA (+) and FL (+) (Figure 1) (Table 1).

### 2.2. Methods

Age, gender, BMI, body fat percentage, abdominal circumference, alanine aminotransferase (ALT), aspartate transaminase (AST), γ-glutamyl transferase (γ-GTP), total bilirubin, albumin, platelets, cholinesterase, total cholesterol, triglyceride, LDL-C, fasting blood glucose, uric acid, and HbA1c levels were compared among the four groups. The presence or absence of FL was diagnosed via abdominal ultrasound using hepatorenal contrast, vascular obscureness, and deep attenuation.

VFA was observed via computed tomography (CT) cross-sectional scans at the umbilical level using a previously reported method [16]. VFA is defined as a cross-sectional area of ≥100 m^2^. The abdominal circumference corresponding to the cross-sectional area is >85 cm in men and >90 cm in women. The amount of VFA is >85 cm in men and >90 cm in women, with an abdominal girth corresponding to >100 m^2^ of the visceral fat area on CT [16]. The body fat percentage was assessed using the SYNAPSE VINCENT system (FJI, Japan). The FIB4 index was calculated as described previously: age (years) × AST (U/L)/platelet count (10^9^/L) × √ALT (U/L) [7,17,18]. Liver fibrosis was classified into FIB4 indices of <1.3, 1.3–2.66, and ≥2.67, where an FIB4 index of <1.3 relates to no fibrosis progression, whereas an index of ≥2.67 relates to possible advanced fibrosis.

The study protocol conformed to the 1975 Helsinki Declaration and was approved by the Institutional Research Ethics Committee (Admission No: 3027).

### 2.3. Statistical Analysis

Statistical analyses were performed using the JMP software (JMP Pro version 13.2 for Windows, SAS, USA), and a *p*-value of <0.05 was considered significant. Baseline continuous variables were expressed as mean ± standard deviation (SD). Between-group comparisons were performed using the chi-squared test for categorical variables and the Shapiro–Wilk test for continuous variables. 

## 3. Results

Among the 663 patients without a drinking history and a BMI of <25 kg/m^2^, 114 (17%) had VFA and/or FL. Among the four groups, group A comprised 549 (control), group B comprised 32, group C comprised 58, and group D comprised 24 patients (Figure 1). The cohort had a slightly higher number of women [299 males (45%) and 364 females (55%)].

### 3.1. Characteristics 

Among the four groups, the group A comprised 212 males (38.6%) and 337 females (61.4%), group B comprised 28 males (87.5%) and 4 females (12.5%), group C comprised 36 males (62%) and 22 females (38%), and group D comprised 23 males (95.8%) and 1 female (4.2%). The average ages for groups A, B, C, and D were 48.5 ± 10.9, 48.8 ± 10.9, 52.7 ± 11.1, and 55.1 ± 12.4 years, respectively. Compared with groups A and B, groups C and D had a higher age. BMI and abdominal circumference were significantly higher in groups B, C, and D than in group A. In particular, group D had the highest abdominal circumference among males and females, followed by groups B, C, and A (*p* < 0.05). The body fat percentage did not differ between the four groups (Table 2).

### 3.2. Complications 

Diabetic complications were observed in 8 (1.5%), 0 (0%), 11 (19%), and 6 (25%) patients in groups A, B, C, and D, respectively. They were significantly higher in groups C and D than in groups A and B (*p* < 0.05). The number of patients with DM was similar between groups A and B. Dyslipidemia complications were observed in 152 (38%), 17 (53%), 31 (53%), and 12 (50%) patients in groups A, B, C, and D, respectively. The number of patients with dyslipidemia was significantly higher in groups B, C, and D than in group A (*p* < 0.05). However, no differences in combined dyslipidemia were observed in groups B, C, and D. A total of 61 patients (12%) in group A, 6 (18%) in group B, 15 (25%) in group C, and 6 (25%) in group D had hypertension. The number of patients with hypertension was higher in groups C and D than in group A (control) (*p* < 0.05), whereas it was the same between groups A and B (Table 3).

### 3.3. Blood Biochemical Examination 

ALT levels were 17 ± 10, 22 ± 9, 28 ± 14, and 36 ± 21 IU/L in groups A, B, C, and D, respectively. ALT levels were higher in groups B, C, D than in group A, particularly in patients with FL (groups C and D) than in those without FL (groups A and B) (*p* < 0.05). AST levels were 20 ± 7, 22 ± 9, 24 ± 6.2, and 26 ± 11 IU/L in groups A, B, C, and D, respectively. γ-GTP levels were 25 ± 24, 50 ± 43, 35 ± 20, and 64 ± 79 IU/L in groups A, B, C, and D, respectively. AST and γ-GTP levels were significantly higher in groups B, C, and D than in group A (*p* < 0.05).

Cholinesterase levels were higher in groups C and D than in groups A and B (*p* < 0.05). Triglyceride levels were higher in groups C and D than in group A and B. Hyperuricemia and fasting blood sugar levels were higher in the patients in groups B, C, and D than in those in group A (control) (*p* < 0.05). HbA1c levels were 5.6% ± 0.4%, 5.6% ± 0.2%, 6.05 ± 0.6%, and 6.0% ± 0.6% in groups A, B, C, and D, respectively, and were significantly higher in groups C and D than in groups A and B (*p* < 0.05). HbA1c levels were similar between groups A and B (Figure 2). Total bilirubin, total cholesterol, and LDL-C levels and platelet count were not significantly different among the four groups (Table 4).

### 3.4. FIB4 Index 

The FIB4 index was 0.9 ± 0.05, 1.0 ± 0.3, 1.4 ± 0.7, and 1.8 ± 0.8 in groups A, B, C, and D, respectively. It was significantly higher in groups C and D than in groups A and B (*p* < 0.05). Furthermore, the FIB4 index was divided into the following three groups: <1.3, 1.3–2.66, and ≥2.67.

Group A had 452, 93 and 4 patients; group B had 27, 5 and 0 patients; group C had 33, 20 and 5 patients; and group D had 5, 16 and 3 patients with FIB4 indices of <1.3, 1.3–2.66, and ≥2.67, respectively.

There were five patients in group C and three patients in group D with FL and an FIB4 index of ≥2.67, which is a marker of liver fibrosis. On the other hand, those in groups A (*n* = 452) and B (*n* = 27) without FL had an FIB4 index of <1.3, accounting for the majority of cases (Figure 3) (Table 5).

## 4. Discussion

In recent years, approximately 1 billion patients worldwide have been diagnosed with NAFLD. Obesity is highly associated with NAFLD, and FL increases the degree of obesity [1]. A total of 15–30% Asians, including the Japanese, with a BMI of 23–24 kg/m^2^ have FL. Conversely, in Westerners, a study has shown that 20–30% of patients with a BMI of 27–32 kg/m^2^ have FL, indicating that Asians have FL with lower BMI values than Westerners [6]. As Asians do not have enough fat cells in their subcutaneous adipose tissues, fat cells become overloaded and begin to overflow as free fatty acids, resulting in lipid accumulation in the visceral adipose tissue, muscle, and liver [18,19]. Therefore, the mechanisms leading to insulin resistance have been studied.

Recently, the subset of individuals with “lean NAFLD” or “non-obese NAFLD,” has become increasingly prevalent. In a meta-analysis using 84 studies (*n* = 10 530 308) for prevalence analysis and 5 studies (*n* = 9121) for incidence analysis from 24 countries or regions, the prevalence of lean NAFLD was 40.8% [20]. NAFLD in lean patients appears to be more common in Asians [21,22,23]. Lean NAFLD is known to cause severe organ damage as the degree of obesity increases [23,24]. In fact, individuals with lean NAFLD have a lower prevalence of metabolic syndrome than those with obese NAFLD; however, they have a higher prevalence of hypertension, dyslipidemia, and type 2 DM complications and higher insulin resistance than healthy people [9,10,21]. A multi-ethnic lean NAFLD cohort from Asia and Italy revealed a slower progression of liver disease in individuals with lean NAFLD [9,10,11,23,24,25,26,27]. Conversely, lean NAFLD accounted for approximately one-third of the individuals with NAFLD (and even higher in older, male, and foreign-born individuals) in a retrospective US study and had higher mortality rates than those with obese NAFLD [28]. Although the pathophysiological mechanisms underlying lean NAFLD have not been elucidated completely, it is reportedly caused by differences in visceral obesity and adipocyte differentiation, changes in lipid turnover, loss of muscle mass, genetic background such as patatin-like phospholipase domain-containing protein 3 (PNPLA3) C > G polymorphism, and different patterns of gut microflora [9].

A BMI of 23–25 kg/m^2^ is also important because it causes metabolic disorders in Asians. There are certain reports of normal BMI in Asian patients with visceral obesity but there are no clinical reports. This is the first report to examine the problem of fat accumulation sites in normal BMI cases in a general screening. We examined the relationship between groups A, B, C, and D and blood tests, metabolic syndrome, and liver fibrosis and verified their effects on the body of non-obese patients who underwent the medical checkup. Among the 663 patients without a drinking history and with a BMI of <25 kg/m^2^ who underwent abdominal ultrasound in this study, VFA or FL or both were observed in as many as 114 patients (17%). Conversely, a total of 549 patients had neither VFA nor FL (83%). Surprisingly, groups C and D had DM complications, higher HbA1c levels, and a higher FIB4 index than groups A and B. The FIB4 index is widely used as an indicator of liver fibrosis in patients with NASH and chronic hepatitis C [7,17,18], indicating that the presence of FL is associated with liver fibrosis. The progression of liver fibrosis in patients with NAFLD is associated with diabetes, hepatocarcinogenesis, and cardiovascular events and is important for determining prognosis [28,29,30,31,32]. Therefore, the presence of FL is extremely important. Conversely, in patients with a BMI of <25 kg/m^2^, the presence of VFA only was not a problem.

There are other papers regarding the importance of FL. In a prospective cohort analysis of 1647 Japanese individuals, the presence of NAFLD was associated with cardiovascular disease (CVD) incidence of 0.6% in non-overweight without NAFLD, 8.8% in non-overweight with NAFLD, in 1.8% in overweight without NAFLD, and 3.3% in overweight with NAFLD. This study concluded that even non-overweight individuals need to be aware of NAFLD to prevent CVD events [33].

Kadowaki et al. [14] reported that in non-obese Japanese men without diabetes, FL without VFA caused insulin resistance in the adipose tissue and muscle as well as a lower hepatic insulin sensitivity; in contrast, patients with VFA but without FL (controls) showed similar insulin sensitivity in the muscle, liver, and adipose tissue. Patients with both VFA and FL had more body fat than the other groups but had similar insulin resistance as those with FL only. Therefore, they reported that the presence of FL was important because it causes insulin resistance in the adipose tissue and muscle and that VFA alone does not cause insulin resistance. Taken together, it is speculated that FL but not VFA alone is a good marker to predict insulin resistance. The results of the report will be considered to support our report. In non-obese people, FL is more likely to cause liver fibrosis and result in a higher frequency of patients with diabetes, according to our study. In addition, it may increase the frequency of the occurrence of CVD [33]. Therefore, the presence of FL can be an important factor for patient prognosis.

Although many obese people often have FL, non-obese patients often have no FL; in fact, the presence or absence of FL cannot be diagnosed without performing imaging examinations, such as abdominal ultrasonography. Therefore, in patients with high cholinesterase, HbA1c levels > 6.0, and DM and particularly in those with an FIB4 index of >2.67, imaging examinations should be performed to check for the presence of FL, even in non-obese individuals.

The current study has some limitations. First, this is a survey at the time of a health checkup and increased visceral fat was judged using abdominal circumference. Increased visceral fat should be judged using imaging examinations, such as CT, in the future. Second, there were few females with high VFA. This may also be because young individuals, including premenopausal females, were included in the medical checkup. In the future, results based on gender should also be considered. Third, this study targeted non-obese patients with a BMI of <25 kg/m^2^ and did not include lean NAFLD individuals with a BMI of <23 kg/m^2^. We hope that future studies use a higher number of patients and perform more expanded studies.

## 5. Conclusions

There is a possibility of visceral obesity even in patients with a BMI of <25 kg/m^2^; among them, FL deposition rather than visceral fat deposition is associated with DM complications and liver fibrosis progression and may be important because it may be related to prognosis. These data suggest that FL, but not VFA alone, is a risk factor of DM and liver fibrosis in non-obese Japanese subjects.

## Figures and Tables

**Figure 1 life-10-00175-f001:**
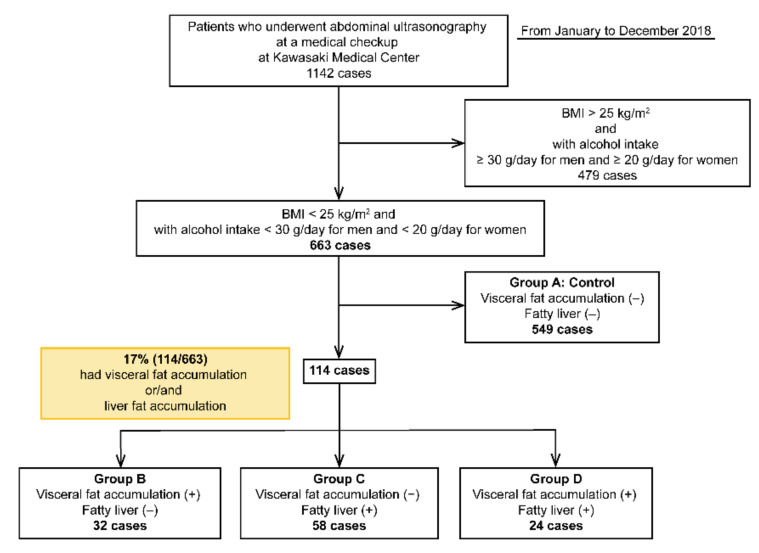
Flowchart of the four groups of patients included in this study.

**Figure 2 life-10-00175-f002:**
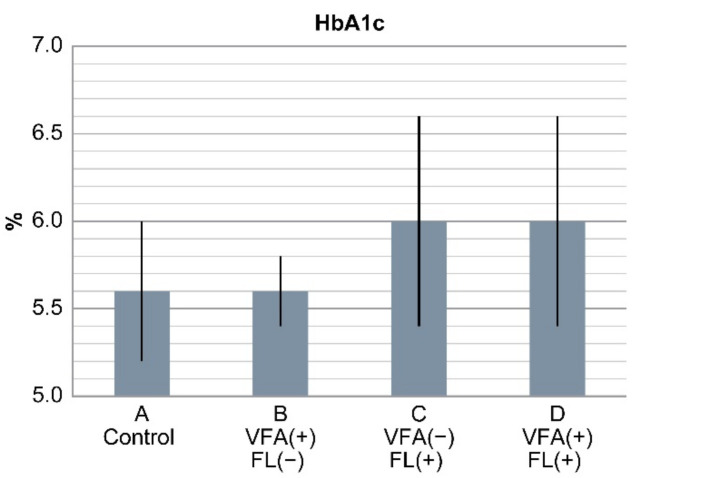
Results of HbA1c levels in the four groups. HbA1c levels were significantly higher in groups C and D with fatty liver than in groups A and B (*p* < 0.0001). HbA1c levels between groups A and B with VFA (+) only were similar. In non-obese Japanese individuals, fatty liver is a risk factor of diabetes, but VFA alone is not a risk factor.

**Figure 3 life-10-00175-f003:**
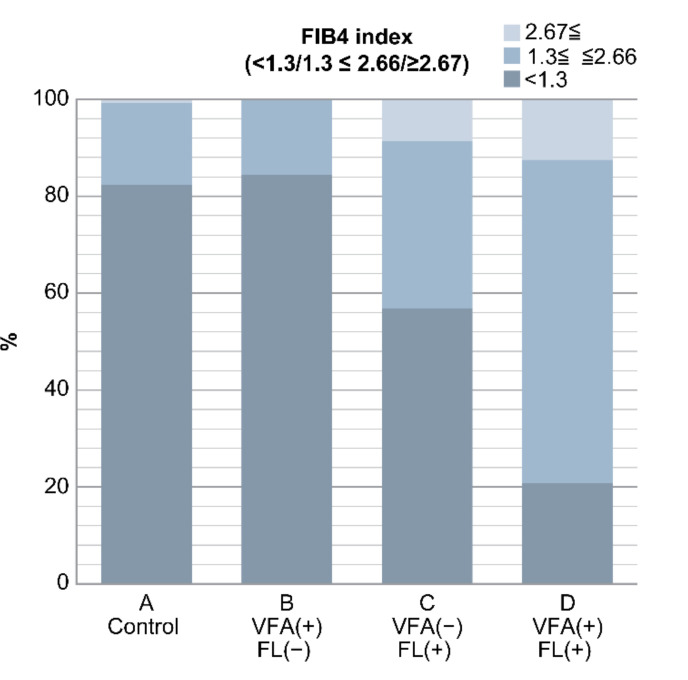
Result of FIB4 index in the four groups. In groups C and D with fatty liver, there were many cases with FIB4 index ≥ 2.67, which is suspicious of liver fibrosis. Conversely, cases with FIB4 index < 1.3 were found in groups A and B without fatty liver (*p* < 0.0001). In non-obese Japanese individuals, fatty liver is a risk factor of liver fibrosis, but VFA alone is not a risk factor.

**Table 1 life-10-00175-t001:** Clinical characteristics of all eligible patients.

Number of Patients	663
Age (years)	50 (22–84)
Gender (M/F)	299/364
BMI (kg/m^2^)	21.5 (13–24.9)
Body fat percentage (%)	23.4 (6.6–39.8)
Abdominal circumference (cm)	
Men (*n* = 299)	80.4 (63.8–96.8)
Women (*n* = 364)	75.8 (58.2–93.5)
Diabetes (+/−)	25/638
Dyslipidemia (+/−)	212/451
Hypertension (+/−)	88/575
ALT (IU/L)	16 (4–132)
AST (IU/L)	20 (9–119)
Γ-GTP (IU/L)	19 (6–338)
T-Bilirubin (mg/dL)	0.9 (0.4–3.2)
Albumin (g/dL)	4.5 (3.2–5.4)
Platelet count (10^4^/µg)	20.6 (9.4–40.1)
Cholinesterase (IU/L)	327 (164–562)
Total cholesterol (ng/dL)	207 (125–316)
Triglyceride (ng/dL)	73 (21–958)
LDL-C (ng/dL)	118 (32–207)
Uric acid	5.0 (1.9–9.4)
Fasting blood sugar	96 (71–296)
HbA1c (%)	5.6 (4.9–12.2)
FIB4 index	0.9 (0.17–5.56)
FIB4 index (<1.3/1.3–2.66/ ≥2.67)	517/134/12

Data are expressed as mean ± SD.

**Table 2 life-10-00175-t002:** Characteristics of the four patient groups in this study.

	Group A Control	Group B VFA (+) FL (−)	Group C VFA (−) FL (+)	Group D VFA (+) FL (+)	*p*-Value
Number of patients	549	32	58	24	
Age (years)	48.5 ± 10.9	48.8 ± 10.9	52.7 ± 11.1 ^b^	55.1 ± 12.4 ^c^	0.0026
Gender, male (%)	38.6	87.5 ^a^	62 ^b,d^	95.8 ^c,f^	<0.0001
BMI (kg/m^2^)	20.9 ± 2.2	23.3 ± 1.0 ^a^	23.1 ± 1.3 ^b^	23.9 ± 0.8 ^c,e^	<0.0001
Body fat percentage (%)	24.0 ± 6.7	25.4 ± 5	26.4 ± 6.7	23.4 ± 3.1	0.0543
Abdominal circumference (cm)					
Men (*n* = 299) (212/28/36/23)	78.0 ± 4.9	87.6 ± 1.9 ^a^	83.2 ± 3.9 ^b,d^	88.7 ± 3.2 ^c,f^	<0.0001
Women (*n* = 364) (337/4/22/1)	74.8 ± 7.6	89.0 ± 2.7 ^a^	82.2 ± 5.0 ^b,d^	93.5	<0.0001

Data are expressed as mean ± SD. VFA: visceral fat accumulation; FL: fatty liver. ^a^
*p* < 0.05 for the chi-squared or Shapiro–Wilk test; group A vs. group B. ^b^
*p* < 0.05 for the chi-squared or Shapiro–Wilk test; group A vs. group C. ^c^
*p* < 0.05 for the chi-squared or Shapiro–Wilk test; group A vs. group D. ^d^
*p* < 0.05 for the chi-squared or Shapiro–Wilk test; group B vs. group C. ^e^
*p* < 0.05 for the chi-squared or Shapiro–Wilk test; group B vs. group D. ^f^
*p* < 0.05 for the chi-squared or Shapiro–Wilk test; group C vs. group D.

**Table 3 life-10-00175-t003:** Complications of the four patient groups in this study.

	Group A Control	Group B VFA (+) FL (-)	Group C VFA (-) FL (+)	Group D VFA (+) FL (+)	*p*-Value
Number of patients	549	32	58	24	
Diabetes (+/−)	8/541 (1.5%)	0/32 (0%)	11/47 (19%) ^b,d^	6/18 (25%) ^c,e^	<0.0001
Dyslipidemia (+/−)	152/397 (38%)	17/15 (53%) ^a^	31/27 (53%) ^b^	12/12 (50%) ^c^	<0.0001
Hypertension (+/−)	61/488 (12%)	6/26 (18%)	15/43 (25%) ^b^	6/18 (25%) ^c^	0.0074

Data are expressed as mean ± SD. VFA: visceral fat accumulation; FL: fatty liver. ^a^
*p* < 0.05 for the chi-squared or Shapiro–Wilk test; group A vs. group B. ^b^
*p* < 0.05 for the chi-squared or Shapiro–Wilk test; group A vs. group C. ^c^
*p* < 0.05 for the chi-squared test or Shapiro–Wilk test; group A vs. group D. ^d^
*p* < 0.05 for the chi-squared or Shapiro–Wilk test; group B vs. group C. ^e^
*p* < 0.05 for the chi-squared or Shapiro–Wilk test; group B vs. group D.

**Table 4 life-10-00175-t004:** Blood biochemical examination of the four patient groups in this study.

	Group A Control	Group B VFA (+) FL (−)	Group C VFA (−) FL (+)	Group D VFA (+) FL (+)	*p*-Value
Number of subjects	549	32	58	24	
ALT (IU/L)	17 ± 10	22 ± 9 ^a^	28 ± 14 ^b,d^	36 ± 21 ^c,e^	<0.0001
AST (IU/L)	20 ± 7	22 ± 9 ^a^	24 ± 6.2 ^b^	26 ± 11 ^c,f^	<0.0001
Γ-GTP (IU/L)	25 ± 24	50 ± 43 ^a^	35 ± 20 ^b,d^	64 ± 79 ^c^	<0.0001
T-Bilirubin (mg/dL)	1.0 ± 0.4	1.1 ± 0.5	0.9 ± 0.4	1.0 ± 0.3	0.1599
Albumin (g/dL)	4.5 ± 0.3	4.5 ± 0.3	4.6 ± 0.2	4.5 ± 0.3	0.0163
Platelet count (10^4^/µg)	21 ± 4.8	21 ± 4.6	22 ± 4.8	21 ± 4.7	0.2361
Cholinesterase (IU/L)	327 ± 68	348 ± 59	394 ± 68 ^b,d^	380 ± 56 ^c,e^	<0.0001
Total cholesterol (ng/dL)	208 ± 34	208 ± 32	212 ± 35	199 ± 32	0.6883
Triglyceride (ng/dL)	79 ± 44	114 ± 93 ^a^	145 ± 123 ^b,d^	138 ± 111 ^c,e^	<0.0001
LDL-C (ng/dL)	119 ± 23	126 ± 31	125 ± 29	117 ± 28	0.1247
Uric acid	5.0 ± 1.3	6.0 ± 1.1 ^a^	5.7 ± 1.1 ^b^	6.1 ± 1.2 ^c^	<0.0001
Fasting blood sugar	96 ± 13	99 ± 7.9 ^a^	106 ± 15 ^b^	105 ± 16 ^c^	<0.0001
HbA1c (%)	5.6 ± 0.4	5.6 ± 0.2	6.0 ± 0.6 ^b,d^	6.0 ± 0.6 ^c,e^	<0.0001

Data are expressed as mean ± SD. VFA: visceral fat accumulation; FL: fatty liver. ^a^
*p* < 0.05 for the chi-squared or Shapiro–Wilk test; group A vs. group B. ^b^
*p* < 0.05 for the chi-squared or Shapiro–Wilk test; group A vs. group C. *c*
*p* < 0.05 for the chi-squared or Shapiro–Wilk test; group A vs. group D. *d*
*p* < 0.05 for the chi-squared or Shapiro–Wilk test; group B vs. group C. ^e^
*p* < 0.05 for the chi-squared or Shapiro–Wilk test; group B vs. group D. ^f^
*p* < 0.05 for the chi-squared or Shapiro–Wilk test; group C vs. group D.

**Table 5 life-10-00175-t005:** FIB4 index of the four patient groups in this study.

	Group A Control	Group B VFA (+) FL (−)	Group C VFA (−) FL (+)	Group D VFA (+) FL (+)	*p*-Value
Number of patients	549	32	58	24	
FIB4 index	0.9 ± 0.05	1.0 ± 0.3	1.4 ± 0.7 ^b,d^	1.8 ± 0.8 ^c,e,f^	<0.0001
FIB4 index (<1.3/1.3–2.66/ ≥ 2.67)	452/93/4	27/5/0	33/20/5 ^b,d^	5/16/3 ^c,e,f^	<0.0001

Data are expressed as mean ± SD. VFA: visceral fat accumulation; FL: fatty liver. ^b^
*p* < 0.05 for the chi-squared or Shapiro–Wilk test; group A vs. group C. ^c^
*p* < 0.05 for the chi-squared or Shapiro–Wilk test; group A vs. group D. ^d^
*p* < 0.05 for the chi-squared or Shapiro–Wilk test; group B vs. group C. ^e^
*p* < 0.05 for the chi-squared or Shapiro–Wilk test; group B vs. group D. ^f^
*p* < 0.05 for the chi-squared or Shapiro–Wilk test; group C vs. group D.

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
