# Peer review of "Fatty Liver, and Not Visceral Fat, Is More Associated with Liver Fibrosis and Diabetes in Non-Obese Japanese Individuals: A Cross-Sectional Study"

_life, 2020, doi:10.3390/life10090175_

Round 1

Reviewer 1 Report

1) Introduction: mention the currently coined term MAFLD

2)METHODS:

  • 2a) provide more details on the patient-flow.
  • - the cohort seems to be skewed toward females
  • - what was the reason patients were seeking healthcare services, what was the indication for ultrasound exam, and for quite extensive laboratory exam?
  • - This is also important as concerns ethical approval: opt-out from what, how was the option written in their medical records, etc. If they were enrolled for the purpose of this study, I am afraid, ethical approval and written consent would be necessary. If the study has analysed dataset collected for other reasons, provide more information.
  • also, it would be of much help for readers to portrait the cohort in which you are up to make important conclusions

- in the methods provide supporting reference for chosen "N"AFLD thresholds 20/30g of ethanol for females/males (known but discussed as too high for Westerners) specifically for Japanese population as there are known differences in alcohol metabolism. In discussion elaborate on this in more detail - it might well be that 30g of ethanol daily is dangerous drinking for livers of patients with "Japanese-type" of alcohol metabolism.

Author Response

We thank the reviewers for their careful review of the manuscript and appreciate their constructive comments. We have revised the manuscript to address each of the issues raised.

For the text, the revised parts are marked in light blue. And, new grammar corrections are marked in yellow.

✔1)Introduction: mention the currently coined term MAFLD

→Thank you for your advice.

In the Introduction section, we have added the term “MAFLD.”

L43: NAFLD was diagnosed by excluding other chronic liver diseases, including “excessive” alcohol consumption. Recently, a positive criterion has been proposed for metabolic dysfunction-associated fatty liver disease (MAFLD). The criteria are based on evidence of hepatic steatosis in addition to one of these criteria: overweight/obesity, type 2 diabetes mellitus, or evidence of metabolic dysregulation

2)Methods

✔the cohort seems to be skewed toward females 

→Thank you for your comment.

We have added the following statement to L125.

L125: The cohort had a slightly higher number of women [299 males (45%) and 364 females (55%)].

✔what was the reason patients were seeking healthcare services, what was the indication for ultrasound exam, and for quite extensive laboratory exam?

provide more details on the patient-flow.

→The target cases included patients who had undergone an annual general medical examination and abdominal ultrasonography for the early diagnosis of health conditions or disease. In Japan, in addition to general medical examinations, the healthcare system permits a detailed medical examination.

We have revised the text as follows (L72):

In total, 1,142 patients underwent abdominal ultrasonography at an annual general medical checkup at Kawasaki Medical Center, Kawasaki Medical School. Annual general medical checkups and abdominal ultrasonography are requested by individuals who desire comprehensive examinations that permit early diagnosis of illness.

✔This is also important as concerns ethical approval: opt-out from what, how was the option written in their medical records, etc. If they were enrolled for the purpose of this study, I am afraid, ethical approval and written consent would be necessary. If the study has analysed dataset collected for other reasons, provide more information.

also, it would be of much help for readers to portrait the cohort in which you are up to make important conclusions

→Thank you for your comment.

We have obtained ethical approval and consent from each patient included in the study.

The use of “opt out” was an error we have corrected it.

We have deleted L68 “An opt-out was used without informed consent,” and added

L115: The study protocol conformed to the 1975 Helsinki Declaration and was approved by the Institutional Research Ethics Committee (Admission No: 3027).

✔in the methods provide supporting reference for chosen "N"AFLD thresholds 20/30g of ethanol for females/males (known but discussed as too high for Westerners) specifically for Japanese population as there are known differences in alcohol metabolism. In discussion elaborate on this in more detail - it might well be that 30g of ethanol daily is dangerous drinking for livers of patients with "Japanese-type" of alcohol metabolism.

→Thank you for your advice.

We believe that causes of the varied effects of alcohol on the human body are multifactorial and include racial and genetic differences. Therefore, this study decided to use alcohol consumption <30 g/day for males and 20 g/day for females, which is the standard guideline for the Japanese Society of Hepatology and for MAFLD.

We have added the following statements to L77: alcohol consumption was determined based on NASH/NAFLD guidelines of the Japanese Society of Hepatology and those of MAFLD.

Reviewer 2 Report

Minor:

Minor:

  1. Lane 52-change to…”insulin resistance in muscle”...Some grammar errors  and typos need to be checked through the text.
  2. Lane 199-.change to .”fat cells” instead of “fat”.
  3. Lane 219-sentence needs to be rewritten since it lacks sense as it is.
  1.  

Author Response

We thank the reviewers for their careful review of the manuscript and appreciate their constructive comments. We have revised the manuscript to address each of the issues raised.

For the text, the revised parts are marked in light blue. And, new grammar corrections are marked in yellow.

Minor:

1  Lane 52-change to…”insulin resistance in muscle”...Some grammar errors  and typos need to be checked through the text.

→Thank you for your advice.

We have revised the text as follows:

L55: Among Japanese men with BMI of 23–25 kg/m2, those with hypertension, hyperglycemia, or dyslipidemia have insulin resistance in the liver but less insulin resistance in the muscle.

2. Lane 199-.change to .”fat cells” instead of “fat”.

→Thank you for your advice.

We have revised “fat” to “fat cells.” (L221)

3 Lane 219-sentence needs to be rewritten since it lacks sense as it is.

→Thank you for your advice.

We have rewritten the causes of lean NAFLD.

L236: Although the pathophysiological mechanisms underlying lean NAFLD have not been elucidated completely, it is reportedly caused by differences in visceral obesity and adipocyte differentiation, changes in lipid turnover, loss of muscle mass, genetic background such as patatin-like phospholipase domain-containing protein 3 (PNPLA3) C > G polymorphism, and different patterns of gut microflora.
